# The Relationship between Christian Religiosity and Adolescent Substance Use in China

**DOI:** 10.3390/ijerph191811233

**Published:** 2022-09-07

**Authors:** Spencer De Li, Jiaqi Lu, Yiyi Chen

**Affiliations:** Department of Sociology, University of Macau, Macau 999078, China

**Keywords:** Christian religiosity, adolescent substance use, China, moral community, Macau

## Abstract

Abundant research has shown that Christian religiosity inhibits adolescent substance use, especially in communities where most of the population shares the same religious values and beliefs. Due to the lack of empirical research, it is unclear if Christian religiosity has the same inhibitory influence in predominantly secular and religiously diverse societies. This study aims to bridge this gap and thereby improve our understanding of the relationship between Christian religiosity and delinquent behavior in different cultural contexts. Through the analysis of survey data collected from a large probability sample of adolescents in China’s special administrative region of Macau, this study found a strong inverse relationship between Christian religiosity and adolescent substance use, despite the predominantly secular nature of Macau society. In contrast, religious commitment among non-Christian youths showed no relationship with substance use. The theoretical and practical implications of the findings are discussed.

## 1. Introduction

In the last several decades, the relationship between the Christian religion and youth development has received growing attention. Abundant research has shown that the Christian religion can promote positive developmental outcomes and inhibit juvenile delinquency, including substance use [1,2,3,4]. Adolescent substance use has been a major public health concern across the world because it can severely impair adolescent physical and psychological wellbeing [5]. Researchers and practitioners have made concerted efforts to identify protective factors that can prevent adolescent substance abuse. The Christian religion has emerged as one of the factors with a potentially strong protective effect. Within this context, the relationship between the Christian religion and substance abuse has become an important topic in child development and health research.

Religious belief and involvement are frequently found to restrain juvenile delinquency, including substance use and abuse. Most of the research attributes this effect to the role of religion as an institution of socialization and social control that fosters social conformity [6,7,8]. Consistent with this explanation, many previous studies have found that religion strengthens social ties [9,10,11] and promotes commitment to conventional beliefs and activities [12,13,14]. It is through these mechanisms that religious commitment has been found to be positively linked to prosocial behavior [15] and negatively correlated with unhealthy behaviors, such as suicide ideation, premature sexual involvement, smoking, drinking, drug use, and delinquency [16,17,18]. However, most of this evidence was established in western societies, where a large segment of the population holds strong Christian identities and beliefs.

Prior research has shown that the Christian faith has the strongest inhibiting effect on delinquent behavior in cultural settings where most of the population is strongly committed to the religious tradition or is actively involved in religious activities organized by Christian churches. However, it is unclear if the same patterns hold true in multicultural settings where the Christian religion is not a dominant social institution. Therefore, additional research is needed to examine if the commitment to the Christian faith has a similar effect on adolescents’ substance use in a cultural setting where Christianity is not a dominant religious institution and does not serve as the foundation of society-wide moral norms.

To provide more evidence on the relationship between religion and adolescent substance use in non-Christian societies, this study examined the potential influence of the Christian faith on substance use among a probability sample of middle school and high school students in Macau, a special administrative region of China. Macau is uniquely suited as a research site for this study because, as a former Portuguese colony, it has a sizeable Christian population. Yet, about two thirds of the general population are not affiliated with any religion [19,20]. Among the known religions, Christianity, including Catholicism and Protestantism, ranks second among the most frequently practiced religions, behind Buddhism. Moreover, Macau is the only place in China where gambling is legal; prostitution is also legal here, although organized prostitution is prohibited. Considering that both gambling and prostitution can act as catalysts for the spread of substance use, the Macau government has recognized substance abuse as a potential major threat to the health and stability of society [21]. In collaboration with non-governmental organizations, the government has developed and implemented many substance misuse prevention and treatment programs. Because of the influence of Western culture, many of these programs are built on the principles of the Christian religion [21]. However, there has been virtually no empirical study examining if the religious tradition has any impact on substance use in the general population or among adolescents in Macau. This study is the first to use data collected from a large probability sample of adolescents to test the relationship between commitment to the Christian faith and substance use in a predominantly secular, but religiously diverse, society.

## 2. Literature Review

### 2.1. Theoretical Background

Christian religiosity is the extent to which an individual is committed to the Christian doctrine that is not only represented by his/her attitudes but also the behaviors that reflect this commitment [19]. An individual is considered to have higher Christian religiosity if they hold strong beliefs in the Christian religion and frequently take part in activities organized by a Christian church [18]. Historically, much of the research on Christian religion and juvenile delinquency has been characterized by a functionalist approach that views collectively held social norms as the key to social conformity [22]. According to this theoretical approach, the Christian religion promotes prosocial behavior by fostering conventional values and beliefs through religious practices. Proponents of the functionalist approach contend that people with higher Christian religiosity are more likely to internalize conventional norms and consider antisocial acts as unacceptable behavior [23,24,25]. Hirschi and Stark, for example, define three ways in which the Christian religion can affect individual behavior: “(1) through its belief system, religion legitimates social and individual values; (2) through its rituals, it reinforces commitment to these values; (3) through its system of eternal standards and punishment, religion helps to ensure the embodiment of values in actual behavior” [25] (pp. 202–203).

Critics argue that the evaluation of the impact of religion merely as a personal trait can lead to a misunderstanding of the influence of religion on individual behavior [26]. The moral community thesis advanced by Stark, Kent & Doyle (1982) contends that the impact of religion on delinquent behavior depends on the extent to which religiosity permeates the culture and social interaction system of the community [27]. In other words, the more pervasive the religious sanctioning system is, the stronger the influence of religiosity. Within this context, the concept of moral community refers to a network of individuals who possess the same religious beliefs and commitment to religious-based behavioral codes [28]. The shared beliefs and experiences of religious individuals contribute to religious homogeneity that serves as a key socializing and social control institution [29]. According to this perspective, personal religiosity is insufficient to prevent adolescents from engaging in delinquent behavior. For Christian religiosity to inhibit delinquent involvement, the adolescent needs to live in a community with a critical mass of other youth or adult believers who share his or her religious beliefs and practices. Indeed, several studies have found that Christian religiosity reduces juvenile delinquency in moral communities, but it has a weak or no effect on antisocial behavior among adolescents in secular communities [30,31,32,33,34]. If the same patterns hold true in Macau, we expect Christian religiosity to have little or no effect on juvenile delinquency because, as a predominately secular society, Macau does not provide the necessary conditions typically observed in a moral community to enable Christian religiosity to perform an effective role in inhibiting delinquency, including adolescent substance use. However, it should be noted that the evidence concerning the moderating effect of moral community on Christian religiosity is inconclusive. While several studies have supported this perspective, much evidence shows that the negative relationship between Christian religiosity and deviance does not necessarily depend on moral communities [35,36,37,38].

### 2.2. Christian Religion and Substance Abuse

Considerable research has shown that the Christian religion inhibits delinquent behavior by providing youth with positive connections with faith communities that produce pro-social values and beliefs [39,40]. Christian religiosity operates as an especially strong protective factor against anti-ascetic actions, which are defined as deviant behaviors that are not clearly proscribed by secular authorities but are in conflict with the tradition of Judeo-Christian support for personal asceticism [41]. Smoking, drinking, and drug use are some of the anti-ascetic actions commonly discouraged or prohibited by Christian churches. For example, the Bible clearly states that “the human body is the temple of the Holy Spirit” [42] (1 Corinthians 6:19–20, ESV) within individuals. Christians must care for themselves by staying away from substance use and other behaviors that might undermine their physical and mental health. Similarly, some Christian scholars insist that “smoking is part of the evil that Satan, as the enemy of all goodness, attempts to spread” and can weaken the bond with the Lord in the mindset of many Catholics [43] (p. 280). Indeed, prior research has found that Bible reading and praying predict negative attitudes toward substance use because adolescents who read the Bible are less likely to build favorable attitudes toward substance use [44]. Additionally, the religious practice of Bible praying helps foster a spiritual awakening that provides individuals with a purpose in life. This has been addressed in the treatment of substance abuse across Western countries. During the treatment programs, the beliefs derived from the practice of praying are supportive of individuals recovering from substance use disorders and maintaining abstinence from substance use [45].

Along with the impact of religious belief, Putnam (2000) demonstrated that Christian religiosity can significantly affect the amount of time juveniles spend in various daily activities [46]. It is common for religious youths to engage in a wide range of prosocial activities well beyond conventional worship, including community, educational, and volunteering activities. Studies commonly assume that participating in faith-based activities acts as a form of displacement where individuals can be directed away from deviancy [47]. For example, attending a Christian church was found to significantly deter substance use because youths with more church attendance were predominantly socialized to resist substance use [17]; little or no church attendance was associated with a more liberal attitude towards substance use [48]. Similarly, an increase in liberal attitudes toward substance use is associated with a decrease in religious involvement [49]. Moreover, involvement in faith-based activities, such as church-sponsored after-school programs, field trips, and athletic programs, could not only enhance adolescents’ interaction and connection with families and peers but also provide them with exposure to adult supervision and structured guidance. Youths who participate more in religious activities with more parental supervision and structured guidance are less likely to engage in unhealthy behavior, including substance use [50].

In sum, much evidence shows that Christian religiosity serves as a protective factor that inhibits adolescent substance use in Western societies, because of the moral values generated by religious rituals and teachings—the shared belief system produces the social integration of morality and norms that inhibit delinquency. Along with the transmission of norms and values, religion also encourages youth to participate in structured and adult-supervised activities that increase their involvement in prosocial behaviors and decrease their time for delinquent acts. Christian religiosity may play a particularly strong role in reducing adolescent involvement in anti-ascetic actions, especially substance use, that are often tolerated by secular authorities but condemned by most Christian churches.

## 3. The Current Study

While studies conducted in the West have produced strong evidence about the impact of Christian religiosity on substance use, additional research is needed to broaden the understanding of the influence of the Christian religion on substance use among adolescents in a secular context. Prior research has shown that Christian religiosity might only have a strong impact on adolescent substance use in moral communities where most of the population is committed to the religious tradition. However, much of the research is based on a comparison of different communities in the West, especially in the U.S. Very little is known about how Christian religiosity is related to adolescent substance use in non-Western societies where Christianity is not a dominant religion. This study aims to narrow this gap by examining the relationship between Christian religiosity and adolescent substance use in Macau, a predominantly secular and religiously diverse society located in southern China. The number of Chinese Christians has increased since the 1980s [51], but, unlike in Western countries, where Christianity is a dominant religion, Christianity has never been a part of the central culture. Christianity is the second-largest religious group in Macau; it coexists with Buddhism, Daoism, Mazuism, Islam, and other religions. Only about 11% of the population in Macau identified themselves as affiliating with either Catholicism or Protestantism, two Christian traditions with a strong presence in China [19]. Judging by the definition provided in previous studies [26], Macau does not meet the criteria of a moral community because of the overall low level of religious affiliation and lack of religious homogeneity in the community. Therefore, Christian religiosity may not have a strong relationship with adolescent substance use. However, researchers are divided over whether moral community serves as a necessary condition for the influence of Christian religiosity on delinquent behavior. A previous study has shown that religious commitment contributes positively to health and wellbeing among the general population of Macau [19]. Despite its lack of permeation in the social and cultural norms, personal commitment to the Christian faith may still serve as a protective factor against adolescent substance use. In this study, we used data collected from a probability sample of adolescents in Macau to investigate the strength and direction of this relationship. We also compared Christian adolescents with non-Christian adolescents to determine if the same patterns of relationship between religiosity and substance use hold for adolescents who do not identify themselves as Christians. Furthermore, by situating the investigation in a predominantly secular society in China, this study provides a partial test of the moral community thesis from an international perspective.

## 4. Methods

### 4.1. Sample and Data

To assess the relationship between Christian religiosity and juvenile substance use, the current study employed data from “Student and Drugs in Macau (SDM)”, a research project funded by the Macau Social Welfare Bureau and conducted by the research team headed by the first author. This survey was designed to assist the government agency investigating substance abuse among secondary school students in Macau and to identify the social and personal factors related to substance use. A stratified cluster systematic probability proportional-to-size sampling method was used in this study to increase the representativeness of the sample. According to the Statistics and Census Service (DESC) of Macau SAR, the region consisted of seven city districts, which served as the stratification factors in the first stage. Schools were randomly sampled from each selected stratum in the second stage. In the third stage, from each sampled school, the team proportionately selected several classes in the seventh, eighth, ninth, tenth, eleventh, and twelfth grades. This three-stage sampling procedure has been demonstrated to be an efficient way of generating probability samples representative of the youth population at the city level [21].

Before survey administration, the research team first arranged a meeting with the officials of each sampled school to review the survey procedure and the content of the questionnaire. After receiving approval from the school officials, the research team set up a survey schedule at the school. On the day of the survey, the research team visited the classes, distributed copies of the questionnaire to the students, and collected the returned surveys. With the assistance of school administrators and teachers, over 3000 adolescents participated in the survey. Applying a listwise deletion procedure, 381 participants were excluded from the analyses because of missing responses on some key variables, which yielded a final sample of 2854 respondents, representing 88% of the total sample of 3235 secondary school students.

To the extent possible, the current survey used standard instruments with demonstrated validity and reliability, including standard questions from Monitoring the Future (MTF), The National Longitudinal Study of Adolescent to Adult Health (Add Health), and the National Youth Risk Behavior Survey (YRBS). The specific instruments adopted in this study are listed in the next section where applicable.

### 4.2. Variables and Measures

The key variables included in this analysis were substance use frequency, religiosity, and Christian religion. We included several control variables that have been identified in prior research as significant correlates of adolescent substance use, including perceived harm of substance use, access to substances, parental monitoring, low self-control, and social bonds [52,53,54,55,56,57,58,59]. We also included demographic measures of birthplace, gender, and grade in the analysis.

*Substance use.* The respondents’ lifetime substance use frequency was used to measure their substance use behaviors. The questions were based on the instruments used in Monitoring the Future (MTF) [60], some of which were modified to fit the local context of Macau. Adolescents were asked, “How many times have you used the following substance”? The specific substances listed under the question included ketamine, ecstasy, heroin, methamphetamine, marijuana, cocaine, codeine, pill, happy water, and others. “Pill” and “happy water” are both new psychoactive substances popular among adolescent substance users in Macau. The sum of the 10 items was calculated, with a higher value indicating more serious problems in substance use. The computed scores range from 0 to 390.

*Christian religiosity.* We used a two-step procedure to measure Christian religiosity. First, we asked the adolescent respondents to identify their religious affiliations. The response categories were “1 = Catholicism”, “2 = Protestantism”, “3 = Buddhism”, “4 = others”, and “5 = no religion”. We computed a dichotomous variable to measure Christian affiliation, with “1” representing Christian religion (self-reported affiliation with Catholicism or Protestantism) and “0” representing non-Christian or no religion (i.e., self-identification with Buddhism, other religions, or no religion). Second, to measure the strength of their religious commitment, we asked the youth respondents three questions drawn from the General Social Survey, including (1) how often they attended religious activities, with response categories ranging from “1 = never attended” to “4 = attended once a week or more”; (2) how important religion was in their lives, with answers ranging from “1 = not important” to “4 = very important”; and (3) how strongly they believed in God, with responses ranging from “1 = I don’t believe in God” to “5 = I know God exists and I have no doubts about it”. The term “God” in the questionnaire refers to the deity, instead of the specific God in the Western culture. These three questions tapped into both the attitudinal and behavioral dimensions of religious commitment [8,18,61,62,63]. Because the response categories had different ranges, the three questions used in measuring religiosity were standardized before further computation. We used the mean score of these three standardized values as the measure of religiosity. It should be noted that we asked all three questions to all respondents in the survey. The answers provided by the adolescents reporting Christian affiliation reflected the strength of their Christian religiosity. Overall, the alpha value computed from the three questions was 0.72, indicating good reliability.

*Perceived harm of substance use.* The measure of perceived harm consisted of 10 attitudinal questions about substance use, including marijuana, cocaine, ketamine, heroin, and methamphetamine. Questions were also adopted from Monitoring the Future (MTF) [60]. Considering that the level of risk attributed to substances varies considerably with the frequency of use, MTF questions were structured to differentiate between occasional and regular use of illicit substances. For each substance, the respondent was asked “How much do you think people risk harming themselves if they use this substance occasionally” and “How much do you think people risk harming themselves if they use this substance regularly”? Respondents rated their attitude to the descriptive substance on a 5-point rating scale that ranged from “1 = no risk for harm” to “5 = serious risk for harm”. We calculated the mean score of the five items, with a higher value indicating a greater perceived risk of harm to the use of the substance. The value of Cronbach’s alpha derived from these items was 0.98, suggesting a very high level of reliability.

*Access to substances.* Access to substances was measured by the sum of several dichotomous variables about the respondents’ access to several substances commonly known in Macau, including ketamine, ecstasy, heroin, methamphetamine, marijuana, and cocaine, as well as unnominated substances in the “other” category. The range of access to substances was from 0 to 7.

*Parental monitoring.* Parental monitoring was measured by a question asking the respondents “how many hours do you spend under parental supervision after school”. The responses were rated on a 6-point scale, ranging from “1 = 0 h” up to “6 = more than 4 h”.

*Low self-control*. The measure of low self-control was obtained by averaging six of the five-item Likert-type scale questions that asked the respondent if “you often lose your temper”, “you often yell and throw things”, “you often quarrel or argue with others”, “you are easily irritated and annoyed”, “you have the urge to hurt or beat people”, and “you have the urge to throw things”. These questions capture the personal traits of impulsivity, physical aggression, and irritability, all of which have been viewed as important dimensions of low self-control [57,64,65]. The Cronbach’s alpha calculated from these three items was 0.86, indicating good reliability. We used the mean scores of these items as the measure of low self-control, with a higher score indicating a lower level of self-control.

*Social bond.* Social bond was measured using a scale developed by Hirschi [66], which is the sum of nine dichotomous variables. This scale has been widely applied in the study of juvenile delinquency and has shown strong test-retest reliability. The scores range from 0 to 9, with higher values indicating stronger bonds to conventional norms.

*Grade*. The participants’ grade was a categorical variable measured by the year of schooling. This study was based on a sample of secondary school students, so the grade ranged from 7 to 12. *Gender.* Gender was a dichotomous variable, with 1 for male and 0 for female. *Birthplace.* Birthplace was also a dichotomous variable, with 1 representing Macau and 0 representing anywhere outside Macau.

### 4.3. Analytic Approach

We first performed a descriptive analysis of the key variables to identify the basic characteristics of the sample. Next, we conducted a multivariate regression analysis to assess the relationship between religiosity and adolescent substance use. “Religiosity × Christianity” was a special variable added to the regression model as an interaction term to assess if Christian religiosity was significantly related to substance use. The interaction term was computed by multiplying religiosity with Christian religion affiliation, which equals 1 when the adolescent reported Christian affiliation and 0 otherwise. The coefficient associated with this interaction term shows how the frequency of substance use changes along with changes in Christian religiosity when all other variables are held constant. Considering the dependent variable is an over-dispersed count variable, we used negative binomial regression to estimate the effects of the independent variables on the dependent variable. Normally, the regression coefficients of the negative binomial model are interpreted as the variance in the logs of the expected counts of the response variable. In the regression analysis, we provided both the regression coefficients and the incidence rate ratios (IRRs) associated with each coefficient. The IRR is the exponential of the coefficient and is interpreted as the change in the incidence rate for every unit increase in the explanatory variable [67].

## 5. Results

### 5.1. Characteristics of the Sample

As shown in Table 1, around fifty-two percent of respondents were male and about seventy-eight percent of them were born in Macau. The grades of the respondents ranged from 7 to 12, with the average grade of the respondents being about 9.45. The mean scores for religious attendance, the importance of religion and belief in God were 1.67, 1.47, and 2.67, respectively. Overall, the scores obtained indicated a moderate level of religiosity. About 13.56% and 10.16% of the respondents self-identified with Christianity and Buddhism, respectively. Less than 1.40% of the respondents self-identified with any other religion. The mean of respondents’ perceived harm from substance use was 4.75, indicating that they believed substance use could cause a great deal of harm to the user. The mean of substance use frequency was 2.72, while the average score for access to substances was only 0.69. Additionally, mean values of 3.61, 2.01 and 3.18 on the measures of parental monitoring, low self-control, and social bond, respectively, were obtained.

### 5.2. Regression Analysis

As hypothesized earlier, Christian religiosity might be related to adolescent substance use. To investigate this possibility, we regressed the frequency of substance use on religiosity, its interaction with the Christian religion, and control variables. As shown in Table 2, the coefficient of religiosity was not statistically significant at the *p* < 0.05 level, indicating that a relationship between religious commitment and substance use, in general, was not observed among Chinese adolescents. The relationship between self-identification with Catholicism and Protestantism (Christian religion) and substance use was positive and significant (β = 2.95, IRR = 19.10, *p* < 0.001). Thus, the incidence rate of using substances among adolescents who self-identified as Christians was much higher than the rate among those who self-identified with other religions or no religion. On the other hand, the coefficient of the interaction term between Christian religion and religiosity was statistically significant (*p* = 0.005), indicating that Christian religion moderated the relationship between religiosity and adolescent substance use. Specifically, Christian religiosity, which was represented by the interaction term when the adolescent reported Christian affiliation, decreased substance use frequency (β = −1.98). It had an IRR of 0.14, meaning that, for every unit of increase in Christian religiosity, the rate of substance use decreased by 86%, while holding all other variables in the model constant.

The results for other important factors were consistent with previous studies. Access to substances was positively and significantly related to substance use (β = 0.44, *p* < 0.01). An IRR of 1.55 suggested that substance use frequency increased by approximately 55% with every one unit of increase in access to substances. Similarly, low self-control was also positively and significantly related to substance use (β = 0.59, *p* < 0.01). According to its IRR of 1.80, adolescent substance use increased by around 80% with every one unit increase in low self-control. Moreover, the perceived harm of substance use (β = −1.44, *p* < 0.001) and social bond (β = −0.23, *p* < 0.05) were both negatively related to adolescent substance use. An IRR of 0.24 suggested that, for every one unit increase in perceived harm of substance use, the incidence rate of substance use decreased by 76%. The IRR for social bonds (0.80) indicated that the incident rate of adolescent substance use decreased by 20% for every unit of increase in social bonds.

A graphic illustration of the relationship between Christian religiosity and adolescent substance use is displayed in Figure 1. Utilizing the regression coefficients in Table 2, we simulated values of Christian religiosity from their minimum to maximum values to estimate the corresponding values of adolescent substance use while holding all other variables constant. The steep slope is consistent with the results in Table 2; it shows that substance use frequency fell precipitously as the value of religiosity increased. Substance use occurred at a very high frequency at the lower end of religiosity. However, substance use dropped significantly close to zero when religiosity moved past a value of 1.5 on the scale and remained relatively stable thereafter. The main effect of religiosity in Table 2 was not statistically significant, hence the moderating effect illustrated in Figure 1 only held for adolescents who identified with the Christian religion. That is, only Christian religiosity demonstrated a strong inverse relationship with substance use.

## 6. Discussion

The objectives of this study were threefold. First, we used data collected from a large sample of Macau adolescents to investigate the strength and direction of the relationship between Christian religiosity and the frequency of substance use. Second, we examined the moderating effect of affiliation with the Christian religion to determine if the pattern of relationship between Christian religiosity and substance use observed among Christian adolescents held for non-Christian adolescents. Third, by conducting the study in a predominantly secular society in China, we sought to address a key question raised by the moral community thesis: is Christian religiosity negatively related to adolescent substance use only in societies where a majority of the population shares the same religious values and beliefs?

The multivariate analysis of the survey data found a strong inverse relationship between Christian religiosity and the frequency of adolescent substance use. Among adolescents identified with the Christian religion, those with the lowest level of religiosity reported a very high frequency of substance use. However, as the level of religiosity increased, the frequency of substance use dropped rapidly. Few adolescents with an above-average level of religiosity reported any substance use. The result indicated that religiosity operated as a protective factor against adolescent substance use in Macau. This protective role of religiosity, however, appeared only to apply to adolescents of Christian faith. Outside of Christian youths, religious commitment did not have a significant relationship with adolescent substance use in Macau. The strong relationship between Christian religiosity and adolescent substance use runs counter to the argument of the moral community perspective that personal commitment to the Christian faith would only inhibit juvenile delinquency in moral communities with a critical mass of believers sharing the same religious beliefs and practices. Compared to societies regarded as “moral communities” in prior research [27], Macau has a much lower relative rate of Christian affiliation among its residents. Less than 14 percent of the population reported being affiliated with the Christian religion. Despite the low rate of affiliation, Christian religiosity could still play a significant role in inhibiting adolescent substance use. Consistent with many previous studies, this study showed that the relationship between Christian religiosity and juvenile delinquency may not be contingent on moral community.

The findings of this study have both theoretical and practical implications. Theoretically, the present study highlights the need to consider that all religions may not play the same protective role against adolescent substance use. The strong relationship between Christian religiosity and adolescent substance use stands in stark contrast to the lack of impact demonstrated by a commitment to other religions. This finding is consistent with prior research suggesting that the Christian religion possesses a unique set of doctrines that, once internalized, can help reduce its believers’ involvement in deviant behavior, especially anti-ascetic actions, such as substance use. Other religions not known for taking such a strong position against substance use may not protect their members from substance use as effectively as the Christian religion. Considering these findings, it would be unreasonable to assume that all forms of religiosity would have the same effect on delinquent behavior. Future studies should include a more in-depth examination of the possibility that religions may influence delinquent behavior in different ways depending on their core teaching and beliefs pertinent to substance use. More studies are also needed to provide further empirical confirmation of the inverse relationship between Christian religiosity and adolescent substance use in a predominantly secular society in a country such as China.

This study also has important implications for the development of effective youth programs for the prevention of substance use in Macau. The government of Macau has been in close collaboration with non-governmental organizations to provide primary and secondary prevention programs to steer adolescents away from substance use. Many of the NGOs are Christian faith-based organizations that promote prosocial behavior through strengthening adolescents’ commitment to Christian doctrines. Until this research, there has been no rigorous empirical study to examine if the commitment to the Christian faith in Macau could reduce adolescent substance use. Through the analysis of the data collected from a large probability sample, this study shows that Christian religiosity can inhibit adolescent substance use among adolescents identified with the Christian religion. This study provides some empirical validation for support of faith-based programs and the model that has been developed to prevent adolescent substance use.

This study has some important limitations that should be taken into consideration when interpreting the results. The data used in the analysis are cross-sectional. The relationships observed in the data are correlational and therefore do not necessarily imply causality. In criminological and health research, it has been a common practice to treat substance use as a function of religiosity in statistical modelling. This approach is not unreasonable, considering that religious affiliation and commitment are often formed before an individual starts using substances in early or middle adolescence. However, such a design is not as rigorous as a prospective study where the time order between religiosity and delinquent behavior can be explicitly determined. To address this limitation, future studies should consider the use of longitudinal data to validate the findings presented in this study. Furthermore, the questions used to measure religiosity might not have the same level of validity across all religious groups. When designing the survey questions, we made a deliberate effort to avoid questions that were applicable only to a specific religion, for example, questions about Bible study or prayers. Despite the effort, the questions might still be more reflective of how religion is practiced in Christianity than in other religious traditions. Future research should pay more attention to the diversity of religious practices and devise more appropriate questions to measure religious commitment in different religions, especially non-Western religions that have not received adequate empirical study to date.

## 7. Conclusions

The present study examined whether Christian religiosity influences adolescent substance use in a predominately secular society. Consistent with the findings from Western societies, our analysis of data collected from a large probability sample of secondary school students showed that commitment to the Christian faith was inversely related to the frequency of substance use among adolescents in Macau. The results indicated that even in a society where religious affiliation of any kind was low, Christian religiosity could still operate as a strong protective factor against adolescent substance use. In contrast, religious commitment was not related to substance use among adolescents who were not affiliated with the Christian religion. The robust relationship between Christian religiosity and adolescent substance use suggests that the Christian religion could inhibit delinquent behavior in cultural settings far beyond the boundaries of moral communities.

## Figures and Tables

**Figure 1 ijerph-19-11233-f001:**
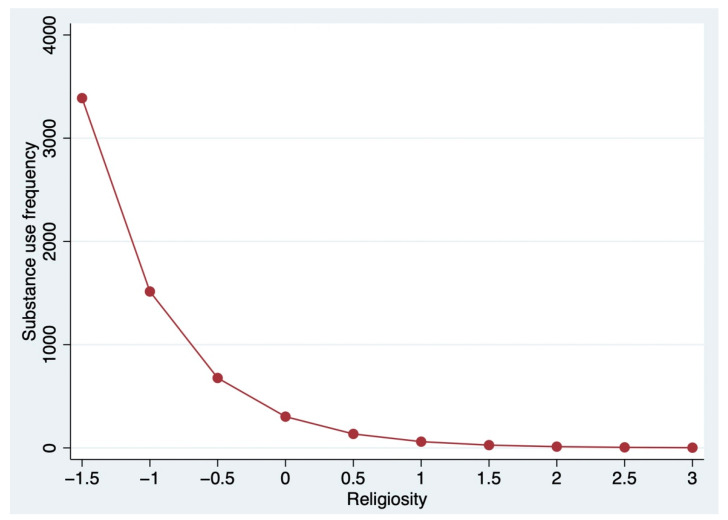
The moderated relationship between religiosity and adolescent substance use frequency by Christianity. Note: The measure of religiosity in the regression analysis is the mean of the standardized scores of the responses to the three questions about religious commitment.

**Table 1 ijerph-19-11233-t001:** Descriptive analysis of key variables (*N* = 2854).

Variables	%/Mean	Std Dev.	Min	Max
Male	51.93%	0.50	0	1
Birthplace (Macau)	77.75%	0.42	0	1
Grade (year)	9.45	1.73	7	12
Christian religion	13.56%	0.34	0	1
Religious attendance	1.67	0.78	1	4
Importance of religion	1.47	0.72	1	4
Belief in God	2.67	1.34	1	5
Access to substances	0.69	1.83	0	7
Perceived harm	4.75	0.63	1	5
Parental monitoring	3.61	1.85	1	6
Low self-control	2.01	0.87	1	5
Social bond	3.18	2.00	0	9
Substance use frequency	2.72	28.28	0	390

**Table 2 ijerph-19-11233-t002:** Negative binomial regression of relationship between religiosity and frequency of adolescent substance use.

Variables	β	SE	IRR	*p*-Value
Grade	0.19	0.12	1.21	0.131
Male	1.69	0.44	5.40	0.000
Birthplace (Macau)	0.41	0.56	1.50	0.464
Parental monitoring	0.04	0.15	1.04	0.778
Access to substances	0.44	0.14	1.55	0.002
Low self-control	0.59	0.22	1.80	0.007
Perceived harm	−1.44	0.37	0.24	0.000
Social bond	−0.23	0.10	0.80	0.023
Religiosity	0.37	0.40	1.45	0.351
Christian religion	2.95	0.77	19.10	0.000
Religiosity × Christianity	−1.98	0.70	0.14	0.005

## Data Availability

The data presented in this study are available on request from the corresponding author.

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
