# Peer review of "The Relationship between Christian Religiosity and Adolescent Substance Use in China"

_ijerph, 2022, doi:10.3390/ijerph191811233_

Round 1

Reviewer 1 Report

This work tries to answer the question of whether in societies with a low religious level the fact that young people holds a Christian religiosity can contribute to inhibit young people from the use of dangerous substances, as drugs. The chosen place is Macau, based on a sample of about 3,000 adolescents. In this Chinese city only 14 percent of the population admits to being Christian. From the empirical data, the authors deduce that the greater the Christian religiosity of young people, the more the use of dangerous substances falls, while in young people with low religiosity or of other religions or who are not religious, the use of these substances increases. What has really surprised me is that this protective role of religion is only considered to apply to adolescents who hold their Christian faith. A priori it seems that this same protection factor could be extended to other religions, such as the Muslim one, for example. More studies of this type would be necessary to confirm this curious conclusion. The authors acknowledge a limitation to their study: The data used in the analysis is cross-sectional. In future studies, the use of longitudinal data should be considered to validate the findings presented in this paper.

In line 464 there is an error: "the Christian region", instead of "the Christian religion"

Author Response

Dear Reviewer,

Many thanks for your time and insightful feedback. Please see the attachment for detailed responses.

Kind regards,

Yiyi

Reviewer 2 Report

Summary

The manuscript, “The Relationship Between Christian Religiosity and Adolescent Substance Use in China” examines substance use among Christian youth in Macau and how it relates to their level of religiosity. The authors’ goal was to determine if religiosity is a protective factor against substance abuse when the predominant religious culture was not the same as the religious affiliation of the adolescents. The authors conducted a survey with 3,022 adolescents from Macau in order to predict their substance use from their religious affiliation and religiosity, while controlling for perceived harm of substance use, access to substances, parental monitoring, self-control, social bonds, grade, and gender. The results of the study indicate that Christian religiosity is a protective factor, despite Macau not being a predominantly Christian cultural context.

Overall Review

Overall, the manuscript has some significant methodological issues to address before it warrants publication. The following issues need to be addressed, as detailed below: the measure of religiosity, the assumptions of the affiliation X religiosity interaction, and the interpretation of the affiliation X religiosity interaction.

Major Issues

Issue #1: Measure of Religiosity

The creation of the religiosity measure has one construct validity problem and one potential external validity problem. Regarding the construct validity problem, three items are averaged to determine “strength of their religious commitment.” However, two of the items are 5-point scale and one of the items was on a 4-point scale. By averaging the responses to these three items, the two 5-point scale questions will contribute more to the final religiosity score than the 4-point scale question. The authors should find another way to create a composite score such that each item contributes equally. Some options might be to include creating a weighted average, standardizing the items first, or using a principal components analysis.

Regarding the potential external validity problem, the three items used are appropriate for assessing religiosity among Christians but may not be appropriate for assessing the religiosity of individuals from other religious affiliations. Given that 86% of the sample is not Christian, the majority of the sample may not have their actual religiosity values appropriately assessed. The authors’ do state, “Answers provided by the adolescents reporting Christian affiliation reflected the strength of their Christian religiosity.” However, if the goal is to only focus on Christian religiosity, there is no point in including the non-Christian sample as a comparison. One item asks participants about their belief in God. However, highly religious Buddhists may not indicate a belief in God. Another item asks about the frequency of attending religious activities. However, some individuals may express their religiosity internally. The authors should look into research on internal and external religious orientation.

One potential solution is to examine each item independently, rather than average them into a single measure. Another potential solution is to only examine Christian participants. Another option is to only look at the item that asks participants about how important religion is in their lives. While this is still far from a perfect externally valid assessment of religiosity, it is less problematic than the other two items.

Issue #2: Assumptions of the Affiliation X Religiosity Interaction

When examining the differences between two subgroups in a sample (i.e., Christian and non-Christian participants), any statistical analysis using the general linear model (which includes linear regression) has certain assumptions. One of those assumptions is that each group has an equal variance. While the general linear model is relatively robust in terms of violations of this assumption, that robustness decreases significantly with unequal sample sizes. The sample sizes in this study are vastly unequal. The authors should conduct and report on an analysis that tests the assumption of equal variance. If the variance is not equivalent between the two groups, then the authors should find a different overall statistical analysis than the linear regression model.

Issue #3: Interpretation of the Affiliation X Religiosity Interaction

In the discussion section, the authors claim that the significant interaction between religious affiliation and religiosity, but insignificant main effect of religiosity is evidence that high religiosity in Christianity is uniquely protective against substance use, even in a cultural context that is not Christian. The statistical analysis does support the claim that high religiosity in Christianity is protective, but not the claim that it is unique as a protective factor. There are two reasons for this. First, is the issue with how religiosity is measured (see above). It is possible that religiosity would be significant for youth from other religious affiliations if a more valid measure of religiosity was used. Second, by grouping youth from all other religious affiliations and non-religiously affiliated youth together, it is impossible to determine if any other affiliation would interact with religiosity. The authors should consider disaggregating the non-Christian category in ways that best fit the sample. The authors could then include more dummy codes to the regression analysis or use a different statistical analysis to look at more than two categories of religious affiliation.

Author Response

(The authors gave the same response as above.)
